# First Indications of Omsk Haemorrhagic Fever Virus beyond Russia

**DOI:** 10.3390/v14040754

**Published:** 2022-04-04

**Authors:** Edith Wagner, Anna Shin, Nur Tukhanova, Nurkeldi Turebekov, Talgat Nurmakhanov, Vitaliy Sutyagin, Almas Berdibekov, Nurbek Maikanov, Ilmars Lezdinsh, Zhanna Shapiyeva, Alexander Shevtsov, Klaus Freimüller, Lukas Peintner, Christina Ehrhardt, Sandra Essbauer

**Affiliations:** 1Section of Experimental Virology, Institute of Medical Microbiology, Jena University Hospital, 07743 Jena, Germany; edithwagner@bundeswehr.org (E.W.); christina.ehrhardt@med.uni-jena.de (C.E.); 2Department of Virology and Intracellular Agents, Bundeswehr Institute of Microbiology, 80937 Munich, Germany; klausfreimueller@bundeswehr.org (K.F.); sandraessbauer@buneswehr.org (S.E.); 3Center for International Health, University Hospital, LMU, 80336 Munich, Germany; annashin86@gmail.com (A.S.); tukhanovanur@gmail.com (N.T.); 4Aikimbayev’s National Scientific Center for Especially Dangerous Infections, Almaty 050000, Kazakhstan; nurik_1976@mail.ru (N.T.); nti0872@gmail.com (T.N.); 5Antiplague Station Taldykorgan, Branch Aikimbayev’s National Scientific Center for Especially Dangerous Infections, Taldykorgan 040000, Kazakhstan; vit197803@mail.ru (V.S.); tpcstald@mail.ru (A.B.); tpcstald.12-zoootdel@mail.ru (I.L.); 6Oral Antiplague Station, Branch Aikimbayev’s National Scientific Center for Especially Dangerous Infections, Oral 090002, Kazakhstan; nmaikanov@mail.ru; 7Scientific Practical Center of Sanitary Epidemiological Expertise and Monitoring, Almaty 050000, Kazakhstan; z.shapiyeva@gmail.com; 8National Center for Biotechnology, Nur-Sultan 010003, Kazakhstan; ncbshevtsov@gmail.com

**Keywords:** Omsk haemorrhagic fever, Republic of Kazakhstan, FUO, ticks, rodents, CSF, flavivirus, tick-borne encephalitis complex

## Abstract

*Omsk haemorrhagic fever virus* (OHFV) is the agent leading to Omsk haemorrhagic fever (OHF), a viral disease currently only known in Western Siberia in Russia. The symptoms include fever, headache, nausea, muscle pain, cough and haemorrhages. The transmission cycle of OHFV is complex. Tick bites or contact with infected small mammals are the main source of infection. The Republic of Kazakhstan is adjacent to the endemic areas of OHFV in Russia and febrile diseases with haemorrhages occur throughout the country—often with unclear aetiology. In this study, we examined human cerebrospinal fluid samples of patients with suspected meningitis or meningoencephalitis with unknown origins for the presence of OHFV RNA. Further, reservoir hosts such as rodents and ticks from four Kazakhstan regions were screened for OHFV RNA to clarify if this virus could be the causative agent for many undiagnosed cases of febrile diseases in humans in Kazakhstan. Out of 130 cerebrospinal fluid samples, two patients (1.53%) originating from Almaty city were positive for OHFV RNA. Screening of tick samples revealed positive pools from different areas in the Akmola region. Of the caught rodents, 1.1% out of 621 were positive for OHFV at four trapping areas from the West Kazakhstan region. In this paper, we present a broad investigation of the spread of OHFV in Kazakhstan in human cerebrospinal fluid samples, rodents and ticks. Our study shows for the first time that OHFV can not only be found in the area of Western Siberia in Russia, but can also be detected up to 1.600 km away in the Almaty region in patients and natural foci.

## 1. Introduction

Omsk haemorrhagic fever (OHF) is a zoonotic infectious disease presently only reported in regions of the Russian oblasts (=regions) Kurgan, Tyumen, Omsk and Novosibirsk in Siberia [1,2,3]. The infection is caused by *Omsk haemorrhagic fever virus* (OHFV), a positive-sensed, single-stranded RNA virus. The virus may infect humans as a dead-end host and can lead to a biphasic course [2]. During the incubation period, infected humans develop unspecific flu-like symptoms followed by the onset of the primary symptoms. These include headache, cough, nausea, chill and muscle pain, but also gastrointestinal symptoms, subconjunctival haemorrhage, nasal, gingival and uterine bleeding, as well as skin haemorrhages accompanied by fever (39–40 °C). Furthermore, the disease leads to extreme sensitivity of the skin (hyperaesthesia) and a petechial rash of the upper body [3,4]. A second phase occurs in 30–50% of patients and leads to encephalitic symptoms such as headache and meningitis, additionally to the primary symptoms. [3]. Patients may suffer from long-term consequences such as weakness, hearing loss, hair loss and mental health problems, in combination with the loss of neurological functions in rare cases [2,5]. The case fatality rate (CFR) of OHF is low (0.5% to 3%) [6], but due to mild cases, with non-febrile patients and only unspecific symptoms without any haemorrhages [3,7], many infections remain undiagnosed.

The first epidemic outbreak occurred in Russia between 1945 and 1958, with 972 confirmed cases and over 1500 suspected cases of OHF. Until 1998, more than 300 OHF cases were confirmed in the regions Tyumen, Kurgan, Omsk and Novosibirsk. Between 1970 and 1975, a study with 577 patients with fever of unknown origin (FUO) showed exposure to OHFV in 18%. The results showed that persons with the mean age of 20–40 years were mainly affected, regardless of sex, as well as 30% of children up to 15 years. Muskrat trappers, poachers and their family members are one of the major risk groups [3,6,8,9,10,11]. No further data about recent outbreaks of OHF in Russia or elsewhere are reported in journals, probably due to non-detected and mild courses [12].

OHFV belongs to the tick-borne encephalitis (TBE) complex of flaviviruses in the family of *Flaviviridae* [13]. Other members of the TBE complex are the *Alkhurma haemorrhagic fever virus, Gadgets Gully virus, Kyasanur Forest disease virus* (KDFV), *Tick-borne encephalitis virus* (TBEV), *Langat virus, Louping ill virus* (LIV), *Powassan virus* (POWV) and the *Royal Farm virus* [3,14,15,16,17]. The high sequence similarity of this group is displayed in the similar morphology, structure and replication mode of the virions. Despite their similar morphology and replication strategy, they cause a variety of clinical symptoms. LIV, POWV and TBEV cause encephalitis, whereas AHFV, KFDV and OHFV mainly cause haemorrhagic fever in humans [13]. Interestingly, OHFV is one of the few viruses that can develop a clinical picture of combined haemorrhages and meningitis and, hence, represents a combination of both groups [2,8,13]. In detail, the 40-nm-large OHFV virion has a spherical or polygonal form, including a nucleolus of 25 nm. The nucleocapsid holds a positive-sense, single-stranded RNA genome wrapped in a capsid protein and is completely enclosed by a host-cell-derived lipid bilayer [11,18,19]. OHFV harbours a genome of 10,787 nucleotides (nt) with an open reading frame (ORF) of 10,242 nt [20]. The ORF is bordered by 5′ and 3′ untranslated regions (UTR). 

The 5′ UTR has a length of 123 nt and contains a secondary structure in a conserved region of a stem loop and an additional stem loop at the 5′ end. However, this additional stem loop was shown to be 29 nt different from other representatives of the TBE complex [9,21]. The 413-nt-long 3′ UTR is shorter compared to other flaviviruses and includes a small variable region and a large core region that is highly conserved. Furthermore, a 3′-poly(A) tail is absent, similar to many other TBEV strains [9,21], and the 3′ UTR is conserved as in other flaviviruses of the TBE complex [9]. The ORF encodes a polyprotein of 3414 amino acids (aa) for three structural proteins (C, prM, E) and seven non-structural proteins (NS1, NS2A, NS2B, NS3, NS4A, NS4B and NS5). This polyprotein is cleaved at conserved cleaving sites during and after the translation process by cellular and viral proteases [9,11]. 

There are contrary opinions on the different subtypes of OHFV and the resulting phylogeny. Currently, there is a discussion on whether two or three subtypes can be distinguished. Early studies from the 1970s suggest that OHFV can be distributed into a minimum of three subtypes [22], which is backed by recent studies from 2015 [23] and 2021 [1]. However, others suggest that the OHFV reference strain *Bogoluvovska* (NC_005062) and the *Kubrin* strain are the same, due to a mix-up of samples in the laboratory [1]. Nevertheless, based on a recent finding, it can be assumed that OHFV circulating in vectors and hosts is very variable and differs markedly from the very early isolated strains [1].

No matter how diverse the OHFV pedigree actually is, the mode of transmission is similar to other members of the genus *Flaviviridae*. It is transmitted to humans by tick bite or via direct contact with infected animals, through respiratory or nutritive pathways. No person-to-person transmission has been reported so far [2,24]. The tick *Dermacentor reticulatus* (termed *D. pictus* in former USSR [25,26]) is the main host for OHFV in the forests and steppes of Siberia [2,24]. In the steppes of Southern and Western Siberia, OHFV is transmitted by *D. marginatus* [27,28]. Both *Dermacentor* ticks act as a parasite on several small mammals and birds in Russia. During their life cycle, they feed on small mammals, ungulates, domestic animals and humans. Risk groups that may be exposed to infected ticks are agricultural workers or individuals picking berries and mushrooms [3]. Further, the tick *Ixodes persulcatus* also plays a role in the life cycle of OHFV [2,29]. Additionally, OHFV was also isolated from mosquitos such as *Coquilletidia richiardii, Oligoryzomys flavescens* and *Ochlerotatus excrucians*, but so far, no data show their role in the ecology of OHFV [30].

OHFV can also be detected in and transmitted by mammals such as the muskrat (*Ondathra zibethicus*) [2]. The muskrat is common in North America, Europe, the Balkans, Russia and Central Asia, including Kazakhstan [31]. The excreta of infected muskrats hold a very high virus load of OHFV [2]. Natural foci of OHFV are typically steppes with lakes and marshes and forests. Many animals are infected by inhalation of the virus contained in aerosols and through contact with or consumption of water that is contaminated by the urine, faeces and/or dead bodies of infected muskrats. These wild hosts contract chronic infections, which sometimes are fatal. OHFV is stable in water for more than two weeks during summer and for more than three months in winter [2,29].

Currently, OHF infections and its agent OFHV are only reported and known in the Russian Federation [1,2]. Thus far, no studies on OHFV in other regions outside of Russia, especially in the neighbouring Republic of Kazakhstan, have been performed. Due to a progressing change in climate and, as a consequence thereof, the changing migration patterns of mammals and birds that may carry ectoparasites such as ticks, the virus can reach other areas than the currently known OHFV endemic areas. 

The Republic of Kazakhstan is located in Central Asia and is adjacent to Russian Western Siberia. It borders China in the east, Turkmenistan in the south and Uzbekistan, Kyrgyzstan, Russia as well as the Caspian Sea in the west. The climate in Kazakhstan has a continental character. Kazakhstan itself is divided into 14 regions with three major cities. It has approximately 19 million inhabitants, with a theoretical population density of seven inhabitants per km^2^—most people, however, are urban citizens [32]. 

In Kazakhstan, there are many cases of fever with unknown origin and many endemic viruses lead to diseases that have similar, non-specific symptoms [33,34,35]. In many cases, the agent cannot be detected because of the complexity and possible cross-reactions due to the close relationship of the flaviviruses within in the TBE complex [9,13,36]. 

We suggest that, due to the close geographic proximity to OHFV-endemic regions in Russia, OHFV might also be present in Kazakhstan. Hence, it should be possible to also detect OHFV in humans as well as reservoir hosts such as rodents and ticks in Kazakhstan. Here, we are conducting for the first time a broad investigation on the spread of OHFV in Kazakhstan by screening patient-derived cerebrospinal fluid samples (CFS) from Almaty, Akmola and East Kazakhstan and hosts such as ticks from North Kazakhstan, Akmola and Almaty and small rodents from the Almaty and West Kazakhstan regions. With these widespread areas, we are able to cover regions that are different in climate and in geographic and potential habitats, to gain more insight into the ecology and complexity of the transmission cycle of OHFV in Kazakhstan.

## 2. Materials and Methods

### 2.1. Human Sample Collection

In the years 2018 and 2019, serous meningitis patients with symptoms such as headache and/or meningeal signs were recruited for a cross-sectional study in the Republic of Kazakhstan in eight hospitals in the regions of East Kazakhstan, Akmola and in Almaty city [37]. After signing an informed consent form, patients’ cerebrospinal fluid samples (CFS) were collected and taken on the day of hospitalisation and stored at −20 °C. This study was performed with the ethical approval of the ethics committees of the Kazakh National Medical University in Almaty, Kazakhstan (opinion number #565) and the Ludwig-Maximilians University in Munich, Germany (opinion number #19-373). 

Each specimen was paired with a detailed, anonymised list of clinical symptoms such as meningitis, encephalitis, headache and/or nausea, vomiting and unconsciousness, as well as a questionnaire with questions regarding sociodemographic factors [37].

### 2.2. Tick Collection

Ticks were collected in three Regions of Kazakhstan (North Kazakhstan, Akmola and Almaty region) at eleven sampling regions, with, in total, 26 different sampling sites during the peak season of tick activity in May and June. In 2016 (for details, see [38]), 2018 and 2019, collections were conducted in the Akmola region in four sampling regions (Sandyktau, Zerendy, Ayrtau and Burabay regions) and in detail in five different villages and surroundings of the villages (villages of Sandyktau, Novonikolskoye, Sadovoye, Imantau and Katarkol). Collections at other sampling sites in the Almaty region, Almaty city and North Kazakhstan were performed in 2018 and 2019. Sampling regions in North Kazakhstan were located in five different areas (Ayirtau, Musrepov, Kyzylzharskiy and Zhumabayev regions, as well as in Petropavlovsk city) in nine different sites (villages of Priozernoye, Novoishymskoye, Anreyevka, Nezhenka, Konyuhovo, city of Petropavlovsk, regions Musrepov, Kyzykzharskiy and Zhumabayev).

The Almaty region was examined in five sampling areas (Alma-Arasan, along the Kapshagay highway, Butakovka, Kapshagay beach and Medeo) and Almaty city itself at seven sampling sites (Baumas Grove, Botanical Garden, grassland areas at the Bridge on Ryskulov and Suyunbay Street and in areas around four bus stations in Almaty city (Appendix A)). 

All ticks were collected by flagging with white cotton sheets and stored in tubes at −20 °C corresponding to the sampling area. Later, ticks were morphologically identified following the official guidelines for tick identification in Kazakhstan [38,39,40]. After identification of the ticks (n_ticks_ = 4993 male and female adults), they were pooled in vials of 1–5 ticks (n_pool_ = 1058) and stored at −20 °C until further analysis.

### 2.3. Rodent Collection

In West Kazakhstan (19 trapping sites in the districts of Bayterek, Borili, Terekti and Taskala), in the Almaty region (city Tekeli, villages Rudnichniy and Bakanas) and in Almaty city, small rodents were trapped with snap traps and pork-fat-covered bread as bait during all seasons of 2018 and 2019 (Appendix A). Captured rodents were visually identified on the species level and, subsequently, the lung and brain tissue from 621 rodents were removed aseptically and stored at −20 °C in RNALater (Thermo Fisher Scientific, Waltham, MA, USA). Rodent trapping was conducted with the ethical approval of the Kazakhstan local ethics committee from the National Scientific Center for Especially Dangerous Infections in Almaty, Kazakhstan (protocol #4, 08.01.18) and the ethical committee of the Ludwig-Maximilians University in Munich, Germany (opinion number 18-631).

### 2.4. Homogenisation and Extraction of Viral RNA

Tick pools, rodent lungs and brain tissues were homogenised with two stainless steel beads and 1 mL of cell culture medium (GibcoTM MEM, Thermo Fisher Scientific, Waltham, MA, USA) using the TissueLyser II (2 min at 30 Hz) (Qiagen, Hilden, Germany). The homogenised samples were centrifuged for 5 min at 2348× *g*. RNA was isolated from 140 µL of the supernatant using a commercial kit (QiAmp Viral RNA Mini Kit; Qiagen, Hilden, Germany), according to the manufacturer’s manual, and stored in aliquots at −80 °C [41].

### 2.5. Real-Time RT-PCR

Real-time reverse transcription (RT)-PCR was used to screen human liquor, tick, rodent lung and rodent brain tissue samples. Primers OHF-d1F (5′-GGCACARACCGTTGTTCTTGAGCT-′3) and OHF-d2R (5′-GCGTTCWGCATTGTTCCAWCCCACCAT-′3) and a TaqMan probe (5′-JOE-AGGTGTTCTGCTGTCTTGTCGAGCACCT-BQH1-′3) were used, detecting a conserved region within the envelope gene of OHFV RNA [2]. The primers bind the region from position 582 to position 720 and produce a fragment of 138 bp in length. The specificity of the used primers and the TaqMan probe was tested against OHFV strain *B**ogolubovka* and other members of the TBE complex, such as *Langat virus*, *TBEV Far Eastern subtype Sofjin* and *LIV.* It was also tested against other flaviviruses, such as *Dengue virus (type 1–4)*, *Zika virus*, *Yellow-fever virus*, *West-Nile virus* (WNV), *Bagaza virus*, *Dakar bat virus*, *Kedougou virus*, *Spondweni virus* and *Usutu virus,* as well as other agents with the potential to cause Encephalitis, such as *Western Equine Encephalitis virus*, *Eastern Equine Encephalitis virus*, *Varicella Zoster virus* and *Herpes Simplex virus (type 1* and *2)* (Appendix A).

### 2.6. Statistical Analysis:

To calculate the prevalence of OHFV in the regions, the Minimum Infection Rate (MIR) method was applied, since the investigated flavivirus only shows a low prevalence (lower than 5%) in the investigated areas [42]. MIR is commonly used to estimate the proportion of infected individuals from pooled samples and is calculated as the ratio of the number of positive pools to the total number of ticks tested. It is assumed that only one infected tick exists in a positive tested pool. Since the arboviruses, which are transmitted via arthropods such as mosquitos and ticks, are relatively rare, the assumption is valid [43]. The MIR was calculated for each sampling site and individually for all tick species. Maps were generated using DIVA GIS [44] and www.mapchart.net (accessed on 16. January 2022).

## 3. Results

In this study, we aimed to identify OHFV, a member of the TBE complex virus family in CFS, in ticks and rodent tissue in the Republic of Kazakhstan. OHFV has a lot of sequence similarity to other closely related flaviviruses. To ensure the specificity of the published primers OHF-d1F and OHF-d2R, and the TaqMan probe targeting the E-gene of OHFV [2], which encodes the envelope protein, we tested them against isolates of several viruses and bacterial agents. The screening panel contained flaviviruses, especially members of the TBE complex, such as *Langat virus* and the *TBEV Far Eastern subtype strain Sofjnn*. Further, we screened for more remotely related flaviviruses including *Dengue virus (type 1–4)*, *Zika virus, Yellow-fever virus* and WNV (Appendix A). The primers did not bind to any of the tested flaviviruses, and also all other tested viruses and bacterial species were negative. Only the OHFV (strain *Bogolubovka*) specimen showed an expected strong signal. Hence, the used real-time RT-PCR assay is highly sensitive and can be exclusively be used for the detection of OHFV.

This establishment of a reliable diagnostic allows for the systematic screening of the existence of OHFV in Kazakhstan. As a proof of principle, we first concentrated on the analysis of a collection of human CFS specimens from patients with symptoms such as headache and meningism [37], being, hence, in the first episode of the typical OHF biphasic clinical course, where a high viral load can be expected. 

Two out of the 130 CSF specimens tested positive for OHFV RNA (1.53%). Both patients were treated in the Almaty City Hospital (Figure 1). All 128 remaining CSF samples were negative for OHFV RNA. 

The first patient was a 33-year-old male, born and living in Almaty city in a rental flat. He stated that he had no contact with any animals such as sheep, cattle, dogs or cats. Furthermore, he did not remember any tick bites. The clinical symptoms that he reported on the day of hospitalisation were headache and neck pain. His temperature, blood pressure and pulse rate were in a normal range. The second patient was a 22-year-old female, born in the Kyzylorda region, but now living in Almaty city in a city apartment. She had recently visited Kyzylorda but also did not remember any contact with animals or any tick bites. She was hospitalised with fever (>37.5 °C) and headache. The blood pressure was in normal boundaries but the pulse rate was increased (<80). 

This detection of OHFV in Kazakhstan inhabitants prompted us to further screen for OHFV in their natural hosts such as ticks and rodents. In total, 4993 ticks from 26 sampling sites from three different regions (Akmola, North Kazakhstan and Almaty region), divided into 1058 pools, were investigated from three regions (Figure 1, light and dark grey). The investigated species were *Dermacentor marginatus* (n_pools_ = 575, n_ticks_ = 2762), *D. reticulatus* (n_pools_ = 456, n_ticks_ = 2167), *D. niveus* (n_pools_ = 6, n_ticks_ = 23) and *Ixodes persulcatus* (n_pools_ = 21, n_ticks_ = 41). In the Sandyktau area (Figure 1), screening of tick samples by using OHFV RT-PCR revealed one positive tick pool of *D. marginatus* from 2016 (1 positive pool out of 90 pools in total (1.1%), MIR of *D. marginatus* at village Sandyktau, Akmola region, in 2016: 0.01) and five positive pools of *D. reticulatus* originating from flagging in 2018 (5 out of 39 pools (12.8%), MIR of *D. reticulatus* at village Sandyktau, Akmola region, in 2018: 0.17). Furthermore, there were two OHFV RNA-positive pools from the Sadovoye area from 2018 (one pool of *D. reticulatus*, and one pool of *I. persulcatus* (2 out of 66 pools (3%), MIR of *D. marginatus* 0.02; MIR of *I. persulcatus* 0.5, both at village Sadovoye in Akmola, 2018). In total, of 21 investigated *I. persulcatus* tick pools, one in Akmola from the Sandyktau area at the village Sadovoye was positive for OHFV (1 out of 21 pools investigated in total, 4.76%) (Table 1). All positive pools originated from the Akmola region (Figure 1, dark grey areas, black triangle). All other tick pools from Akmola 2018 and 2019, North Kazakhstan 2018 and 2019 and the Almaty region 2019 were negative for OHFV-RNA (Figure 1, light grey areas).

The natural targets of ticks are small mammals. Therefore, 621 small rodents from two regions (Almaty and West Kazakhstan, Figure 1, light and dark grey areas) were included in this study in order to learn more about the natural reservoir of OHFV in Kazakhstan. Baiting with pork fat led to the capture of a broad range of rodents, including the family Muridae represented by *Apodemus uralensis* (*n* = 259), *Mus musculus* (*n* = 128) and *Rattus norvegicus* (*n* = 39), the family Gerbilinae represented by *Meriones meridianus* (*n* = 2), the family Cricetidae represented by *Microtus arvalis* (*n* = 86), *Microtus kirgisorum* (*n* = 49) and *Clethrionoyms glareolus* (*n* = 12) and the family Gliridae represented by *Dryomys nitedula* (*n* = 15). Additionally, insectivores such as *Crocidura suaveolens* (*n* = 28), Soricidae (*Sorex araneus* (*n* = 1), *Sorex minutus* (*n* = 1) and a not further classified *Sorex* spp. (*n* = 1)) were trapped and lung and brain tissue samples were screened for OHFV (Appendix A).

In three lungs of rodents of the species *C. glareolus*, captured in the Teretki District in West Kazakhstan in 2018, OHFV RNA was detected (*n* = 3/9, 33.3%) (Table 2). Furthermore, viral OHFV RNA was isolated from two lungs of *M. musculus* from the Taskala District (*n* = 2/16, 12.5%), in one specimen of *A. uralensis* from Oral City (*n* = 1/27, 3.7%), as well as from one specimen of *A. uralensis* from the Bayterek District (*n* = 1/58, 1.7%) (Figure 1). All positive OHFV samples were from the West Kazakhstan region (Figure 1, dark grey area, black circle).

All other lung tissue samples, as well as all brain tissue samples from small mammals originating from the West Kazakhstan region (2019), Almaty region and Almaty city from 2018 and 2019, were negative for OHFV RNA.

Although it was shown that the selected diagnostic primers identified OHFV with high reliability (Appendix A), there remains the possibility of a false positive result. A potential cross-reactivity was ruled out by testing both positive human CFS for other closely related flaviviruses such as WNV and TBEV. Both patients did not have any previous infections with either TBEV or WNV [37]. Similarly, all positive OHFV samples originating from ticks were negative for TBEV (data not shown)*, Crimean-Congo haemorrhagic fever virus* (CCHFV) [42] and *Rickettsia* spp. [45]. All positive OHFV samples originating from rodents were negative for *Orthohantavirus* and *Rickettsia* spp. (data not shown). 

In conclusion, OHFV was, for the first time, detected in human patients, ticks and rodents outside of Russia in the Republic of Kazakhstan.

## 4. Discussion

OHFV is currently only known to be prevalent in four regions of Russia, namely Kurgan, Tyumen, Omsk and Novosibirsk. Only in these regions, irregular but frequently recurring outbreaks of OHF among the population are recorded [1,2,3,7]. The Republic of Kazakhstan is adjacent to the known OHFV-endemic areas in Russia—North Kazakhstan, for example, borders directly the regions of Kurgan, Tyumen and Omsk. The major arthropod vector of OHFV is the tick *D. reticulatus*, which was also the tick mainly collected in the regions of North Kazakhstan and Akmola (Table 1). Furthermore, sightings of the muskrat *(O. zibethicus*), the most well-known spreader of OHFV, have been reported along the rivers of Kazakhstan [30]. Since all these mentioned hosts are also endemic to Kazakhstan, it is actually not a question of if but how widespread OHFV is in Kazakhstan.

For this reason, we examined areas of Kazakhstan that are adjacent to the Russian OHFV epidemic areas, such as North Kazakhstan and the Akmola region. In order to establish a cross-sectional study, it was supported by also examining more remote regions such as West Kazakhstan in the west and East Kazakhstan and the Almaty region in the south east of Kazakhstan. Arthropod-borne viral diseases usually have a natural cycle that includes ticks and small rodents as natural reservoirs. The human is usually a dead-end host. In order to draw a complete picture of the distribution of OHFV in Kazakhstan, we screened CSF from human patients with meningitis, pools of ticks and captured small rodents. 

The screening on CSF was based on a collection of 130 samples from a previous study about serous meningitis patients of unknown aetiology [37]. However, in this study, only for approximately 20% of the enrolled patients was a causative agent, such as TBEV or WNF, identified. Due to the lack of a reliable OHFV ELISA, the screening had to be performed on viral RNA, although the timeframe during which the patient is viraemic is rather short. Nonetheless, two patients with OHFV showing very weak symptoms were identified, interestingly in Almaty city, far away from the presumed endemic areas in the north of Kazakhstan. Both of the patients did not have a travel history to the north, but one of the positive patients had recently travelled to Kyzylorda in the south. As the OHFV virus is shed by infected animals in a very high titre, it is possible that they became infected by drinking, swimming or washing their hands in contaminated water. Hence, a subsequent screening of rodents and ticks in the area of Almaty city was initiated but did not reveal any positive specimen. However, it should be noted that the number of ticks and rodents examined in the area of Almaty was too small to draw a definite conclusion from our results. 

Nevertheless, in other areas, ticks and rodents were indeed positive for OHFV. We identified ticks of the family *Dermacentor* that were positive for OHFV, confirming its important role in the spread of OHFV. Furthermore, we also detected positive pools of *I. persulcatus*. As already discussed elsewhere, *Ixodes* seems to play a potential role in the sylvatic life cycle of the agent [2,29].

Surprisingly, rodents from West Kazakhstan, a region far away from the known OHFV-endemic regions in Siberia, tested positive for OHFV. It was interesting, however, that all positive rodents were collected in areas close to lakes and rivers. The infected rodents of West Kazakhstan, for example, were collected in the village of Teretiki and around the city of Oral, which is close to the river Ural. Additionally, in Akmola, the positive ticks (*D. marginatus* and *D. reticulatus* and also *I. persulcatus*) were collected in forest steppe areas with many lakes and rivers. In contrast to this, all tick and rodent sampling sites in the region of North Kazakhstan were quite distant from the next river. One could postulate that the spread of OHFV follows the tracks of waterways, since the muskrat has populated rivers in Central Asia since its release there in the 1930s [31,46]. By excreting the virus through their urine and faeces in the water, muskrats infect other (semi)-aquatic small mammals such as *Arvicola amphibius* [3,29]. In addition, OHFV infects other small rodents such as members of *Microtus* spp. [2,29]. Blood-feeding ticks enhance the local distribution of OHFV in new endemic areas [2,3]. Additionally, there is a theory that OHFV has existed unnoticed in Siberia, including Kazakhstan, [3,47], for many decades already and started to cause human casualties only by the introduction of the muskrat. This is opposed by a recently published study that states that OHFV has evolved beyond the TBEV Far Eastern subtype, due to the fact that TBEV has spread in the muskrat population without any involvement of ticks—leading to a new zoonotic agent [48].

In general, information is scarce on the effect of a persistent OHFV infection on its natural hosts. One can assume that all OHFV-positive rodents probably had asymptotic infections, since viral RNA was not detected in their brains. However, laboratory tests propose a fatal neuroinfection [49], and in the wild, *O. zibethicus* is particularly susceptible to acute neuroinfection leading to death [29]. However, other wild vertebrates show mostly asymptomatic diseases with viraemia [3,29]. 

The question of how OHFV actually came to Kazakhstan will remain the subject of future studies. Due to the fact that human patients are infected in Almaty city also, which is approximately 2000 km away from West Kazakhstan and 500 km away from the Akmola region, it is possible that OHFV is already widespread throughout Kazakhstan. The spread could also have occurred through bird migration, through which infected ticks can travel great distances. Other possibilities include China’s Belt and Road Initiative, animal trading, and travelling, whose role in the dissemination of ticks and rodents and potential agents should not be underestimated [50,51]. 

However, despite this first description of OHFV in Kazakhstan, there is no urgent need for public health measures. OHF has a rather low CFR (0.5–3%) [6], contrary to other endemic zoonotic infections in Central Asia, such as the TBEV Far Eastern subtype with a CFR of 20–30% [52] or CCHFV with a CFR of approximately 30% [53]. Nonetheless, there is a high chance of subclinical or mild cases with only fever and no haemorrhages. This means that infections may remain undiagnosed and undetected [3,7] and any febrile disease can be declared as FUO, which is a noteworthy public health problem [42]. Recently published studies on FUO [34,42,54,55] and a study on serous meningitis in patients in Kazakhstan demonstrate that, in general, there is a need to raise awareness of the effect of emerging zoonoses in the healthcare system of Kazakhstan. There is a need for more reliable diagnostics and an awareness that zoonoses can potentially lead from an initially unspectacular febrile disease to a life-threatening condition. Currently, laboratory diagnosis for OHFV is based on serological methods including an ELISA against OHFV IgM and IgG antibodies [2] or against the OHFV-NS1 antigen [56]. Further detection is performed by seroconversion with paired sera by haemagglutination inhibition, complement fixation test, and neutralisation assays [2]. These tests should be available in all infection hospitals in Central Asia to enable an efficient differential diagnosis. 

To complete this interesting evidence of an OHFV distribution in Kazakhstan, the positive samples need to be sequenced. It is also not possible to say into which OHFV subtype according to Kovalev et al., 2021 [1] the positive samples from Kazakhstan can be classified. Unfortunately, due to the current pandemic and political situation, the sequencing was not possible in this study. However, future investigations including sequencing will show which strains are circulating in wild host animals with regard to the different clusters that seem to exist [1]. Furthermore, a distance analysis will clarify how long OHFV has existed in the regions or whether it has appeared there only recently.

Therefore, it is important to start a systematic cross-sectional and longitudinal study in Kazakhstan including more data of OHFV in patients, rodents and ticks to clarify open questions and to gain a complete picture of OHFV in Kazakhstan—a virus that now officially has left Russian territory and spreads over Central Asia. Future studies will give a deeper insight into the emergence and spreading mechanisms of OHFV all over Kazakhstan.

## Figures and Tables

**Figure 1 viruses-14-00754-f001:**
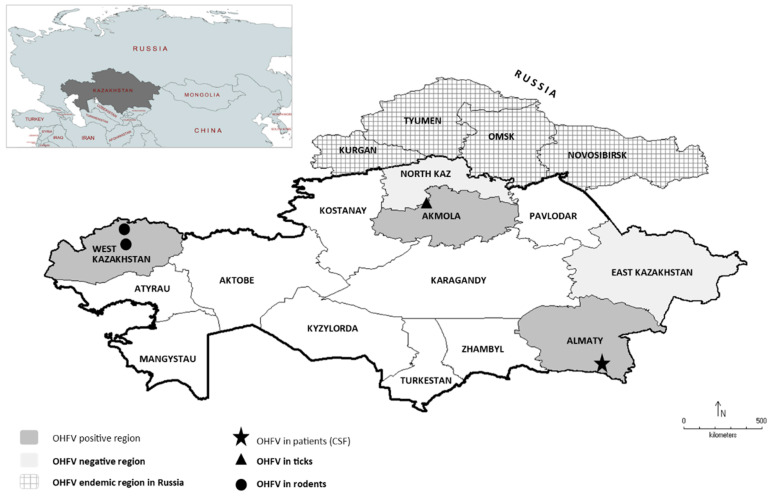
Investigation of Omsk haemorrhagic fever virus (OHFV) in Kazakhstan. Patients with suspected meningitis or meningoencephalitis in East Kazakhstan, Akmola and Almaty city, rodents from Almaty city, Almaty region and West Kazakhstan and ticks originating from North Kazakhstan, Akmola region, Almaty city and Almaty region were screened for OHFV RNA (light and dark grey). OHFV regions identified in this study (dark grey) with OHFV-RNA-positive patients (★), ticks (▲) and rodents (●). In North Kazakhstan and East Kazakhstan, no OHFV was identified (light grey). Thus far, OHFV-endemic regions in Russia adjacent to Kazakhstan are Kurgan, Tyumen, Omsk and Novosibirsk (grey pattern).

**Table 1 viruses-14-00754-t001:** Results of OHFV real-time RT-PCR screen in tick pools of *D. marginatus, D. reticulatus*, *D. niveus* and *I. persulcatus* (in total, 1058 tick pools) from Almaty city (seven trapping sites), Almaty region (five trapping sites), Akmola region (six trapping sites) and North Kazakhstan (nine trapping sites).

Region	Year	Collected Species	Positive Species (#Of Pools)
Almaty region	2018/2019	1	0
Almaty city	2018/2019	2	0
Akmola region	2016/2018/2019	4	*D. marginatus* (2), *D. reticulatus* (6), *I. persulcatus* (1)
North Kazakhstan	2018/2019	4	0

**Table 2 viruses-14-00754-t002:** Results of OHFV real-time RT-PCR screen in lung tissue of small mammals (*n* = 621) from Almaty region (three trapping sites), Almaty city (seven trapping sites) and West Kazakhstan (19 trapping sites) from 2018/2019. In total, eleven species were collected.

Region	Year	Collected Species	Positive Species (#of positive)
Almaty region	2018/2019	6	0
Almaty city	2018/2019	7	0
West Kazakhstan	2018/2019	6	*A. uralensis* (2), *M. musculus* (2), *C. glareolus* (3)

## Data Availability

Not applicable.

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
