# Peer review of "First Indications of Omsk Haemorrhagic Fever Virus beyond Russia"

_viruses, 2022, doi:10.3390/v14040754_

Round 1
Reviewer 1 Report
The authors report the detection of RNA of Omsk haemorrhagic fever virus (OHFV) in clinical samples, ticks, and rodents in Kazakhstan. The manuscript is of great interest and importance because OHFV was previously known only in Russia.
Specific comments:
- the title and further text: I am not very convinced that one can speak of "emergence" in this context. The fact that OFHV RNA was discovered recently does not mean that there is an emergence of the virus, since this virus has not been studied before in this area. In order to speak of an emergence, the number of infected humans/ticks/rodents would have to have increased compared to the previous period.
- Line 32: for the presence of OHFV RNA
- Line 33: were screened for the presence of OHFV RNA
- Line 48: OHF is not an emerging disease because there are neither reports about an increasing number of infected people nor the disease spreading to new areas
- Line 83: infection with KFDV can also cause neurological involvement
- Line 85: nucleolus? – did you mean nucleocapsid?
- Line 95: why only of the Far-Eastern and Siberian TBEV strains? European strains also do not have poly(A) tail (there are only two European strains known to have poly(A) segment in the 3’ UTR, but it is not a poly(A) tail)
- Line 110: it is better to say: similar to other members of the genus Flavivirus. HCV is also a member of the family Flaviviridae, but the mode of transmission is very different.
- Lines 189-190: nticks and npool – ticks and pool in a subscript
- Line 202-203: In most countries, an approval from Natural Preserve authorities is needed for field collection of animals. Please, provide a number of the approval or statement that there was no need for the permission from natural preserve authorities and provide an appropriate justification.
- Line 218: check spelling of the OHFV strain
- Line 219: not Sofyn, but Sofjin in English transliteration
- Lines 218-219: why there was no European TBEV strain included? European TBEV subtype is known to be present in Siberia, so one could expect European strains in Kazakhstan too.
- Line 223: Varicella-zoster virus (check spelling)
- Line 242: Sofjin
- Line 244 and further: the most common abbreviation for West Nile virus is WNV
- Line 246: check spelling of the OHFV strain
- Figure 1: the map is somewhot sqeezed
- Line 268: that he had no contact
- Line 313: M. musculus – italics
- Line 331: Rickettsia spp. (TBEV) – seems to be a mistake
- Major comment: why the PCR fragments were not sequenced? The authors should provide a sequence analysis for the positive samples. First, they could demonstrate that the results are correct, second, they could generate a phylogenetic tree comparing the OHFV strains from Kazakhstan and Russia
- Supplementary Table 3 contains an unacceptable number of typos in the names of viruses (for example, not Varizella zoster virus, but varicella-zoster virus; there is no OHFV strain Bogulokova; not Sofyn, but Sofjin; not Yellwo viver virus, but yellow fever virus; not West-Nil virus, but West Nile virus...).
Author Response
Reviewer 1:
The authors report the detection of RNA of Omsk haemorrhagic fever virus (OHFV) in clinical samples, ticks, and rodents in Kazakhstan. The manuscript is of great interest and importance because OHFV was previously known only in Russia.
Specific comments:
- The title and further text: I am not very convinced that one can speak of "emergence" in this context. The fact that OFHV RNA was discovered recently does not mean that there is an emergence of the virus, since this virus has not been studied before in this area. In order to speak of an emergence, the number of infected humans/ticks/rodents would have to have increased compared to the previous period.
Our response: Thank you for pointing out the alarmist tone of our choice of the headline. We now changed the headline to “First indications of OHFV in Central Asia beyond Russia”.
- Line 32: for the presence of OHFV RNA
Our response: Thank you, proposed change was included.
- Line 33: were screened for the presence of OHFV RNA
Our response: Thank you, proposed change was included.
- Line 48: OHF is not an emerging disease because there are neither reports about an increasing number of infected people nor the disease spreading to new areas
Our response: Thank you, proposed change was included. We excluded the term emerging, although we here show in our paper, that OHFV has to be considered as an emerging or spreading virus.
- Line 83: infection with KFDV can also cause neurological involvement
Our response: Thank you for pointing out this mistake and please excuse our ignorance. We rephrased the sentence to a less totalitarian tone and included an additional reference [1].
- Line 85: nucleolus? – did you mean nucleocapsid?
Our response: Sorry for this mistake – of course we meant the nucleocapsid and we changed the text accordingly.
- Line 95: why only of the Far-Eastern and Siberian TBEV strains? European strains also do not have poly(A) tail (there are only two European strains known to have poly(A) segment in the 3’ UTR, but it is not a poly(A) tail)
Our response: Thank you for pointing out this impreciseness. Since only two of the many TBEV strains (Neudoerfl and a bohemian strain) actually have a polyA tail, we now rephrased the sentence and state “similar to many other TBEV strains”.
- Line 110: it is better to say: similar to other members of the genus Flavivirus. HCV is also a member of the family Flaviviridae, but the mode of transmission is very different.
Our response: Thank you for this comment. We changed the text to “Similar to other members of the genus”.
- Lines 189-190: nticks and npool – ticks and pool in a subscript
Our response: Thank you, proposed change was included.
- Line 202-203: In most countries, an approval from Natural Preserve authorities is needed for field collection of animals. Please, provide a number of the approval or statement that there was no need for the permission from natural preserve authorities and provide an appropriate justification.
Our response: Thank you for this remark. Indeed we first had to ask for permission at the national scientific committee and got permission for conducting our research. Furthermore we got clearance by the ethical committee of the University of Munich/Germany where some of our co-authors are located. We stated the ID numbers of the approval documents in the material and methods section. „Rodent trapping occurred after ethical approval of Kazakhstan local ethics committee at National Scientific Center Especially Dangerous Infectious in Almaty, Kazakhstan (protocol #4, 08.01.18) and the ethical committee of the Ludwig-Maximilian University in Munich, Germany (18-631).” If the reviewer wants to see the original documents (the Kazakhstan document is in Russian only) I am happy to provide them upon request.
- Line 218: check spelling of the OHFV strain
Our response: Thank you, proposed corrections were included. We changed it in line 218 and 246
- Line 219: not Sofyn, but Sofjin in English transliteration
Our response: Thank you, proposed change was included.
- Lines 218-219: why there was no European TBEV strain included? European TBEV subtype is known to be present in Siberia, so one could expect European strains in Kazakhstan too.
Our response: Thank you for this valid comment. Indeed we already performed extensive investigations on the spread of TBEV in Kazakhstan. As shown in one of our previous publications from 2020, in Kazkshtan only Siberian subtype of TBEV was detected in Ticks [2, 3]. That’s why we focussed on Sibirian subtype of TBEV. This is further confirmed by our most recent TBEV investigation in infected human patients. However, of course, the reviewer is right that the European TBEV strain was also reported in Siberia, but this study focussed on an area rather far away from Kazakhstan and not connected by classical e.g. bird migration routes. Hence, we do not expect this European subtype to arrive in Kazakhstan anytime soon.
Line 223: Varicella-zoster virus (check spelling)
Our response: Thank you, proposed change was included.
- Line 242: Sofjin
Our response: Thank you, proposed change was included.
- Line 244 and further: the most common abbreviation for West Nile virus is WNV
Our response: Thank you, proposed changes were included in Lines 221, 244, 328 and 329
- Line 246: check spelling of the OHFV strain
Our response: Thank you, proposed change was included.
- Figure 1: the map is somewhot sqeezed
Our response: Thank you for this comment. Indeed this is a common problem with the depiction of such a huge country as Kazakhstan. In the top left corner we show a map that shows entire Cental Asia including Russia. That’s why one uses the Robinson projection, where Kazakhstan gets stretched in its North-South expansion. In the main image we used an ArcGIS generated map (I think it is based on a Mercator projection). Hence, the shape of Kazakhstan sometimes looks sometimes somewhat disturbed.
- Line 268: that he had no contact
Our response: Thank you, proposed change was included.
- Line 313: M. musculus – italics
Our response: Thank you, proposed change was included.
- Line 331: Rickettsia spp. (TBEV) – seems to be a mistake
Our response: Thank you for pointing out this confusion in the assignment of the quotations. We corrected the Rickettsia spp. to references 45, Crimean-congo to reference 42 and our TBEV study on ticks is unfortunately still far from being submitted.
- Major comment: why the PCR fragments were not sequenced? The authors should provide a sequence analysis for the positive samples. First, they could demonstrate that the results are correct, second, they could generate a phylogenetic tree comparing the OHFV strains from Kazakhstan and Russia
Our response: Thank you for raising this issue! We are aware that verification by sequencing would be the golden standard to prove this results. Unfortunately, due to political and pandemic cirucumstances it was not possible so far to perform sequencing on positive OHFV samples. Kazakhstan legislation does not allow an export of (inactivated) biological material from Kazakhstan and all sequencing needs to be performed in national institutes that got severely affected by the uprisings in January this year. And due to the nationalistic laws, no sequencing is possible outside of Kazakhstan as for instance in Germany.
However, still we would like to publish the current available data to inform the Kazakhstan public health institutions located at the Ministry of Health in Kazkahstan and the Gouvernment of Kazakhstan that OHFV is a possible emerging threat in Kazakshtan that is worthy for further investigation and collaboration on this topic and needs consideration in patient care. We want to assure you that used primer and TaqProbe sequences are highly sensitive and specific for OHFV. We have confirmed it by testing the primers against several flaviviruses and for flaviviruses that are closely related to OHFV.
- Supplementary Table 3 contains an unacceptable number of typos in the names of viruses (for example, not Varizella zoster virus, but varicella-zoster virus; there is no OHFV strain Bogulokova; not Sofyn, but Sofjin; not Yellwo viver virus, but yellow fever virus; not West-Nil virus, but West Nile virus...).
Our response: Thank you for pointing out this embarrassing mistakes. Indeed the supplement slipped through our internal quality control. I am sorry for the extra work for the reviewer.

Reviewer 2 Report
This paper reports identification of Omsk hemorrhagic fever virus (OHFV) outside Siberia in Kazakhstan. This is a potentially important paper as it describes an expanded geographic range for the virus based on identification of OHFV in human, ticks and rodents. However, it is all based on qRT-PCR and no sequencing of PCR products, nor virus isolation. This would be particularly important as it would provide a second method to confirm the PCR results and also potentially look at genetic variation versus OHFV in Russia. To rely on one technique and make big conclusions has difficulties. Thus, at the very least a PCR product should be sequenced to confirm it is OHFV and compared to published OHFV strains.
The qRT-PCR is of concern as it is based on published primers from a book article not an article in a journal, so it is not easy to access it. I could not get hold of the article. The size of the amplicon is not stated and none of the control RT-PCR results are shown to demonstrate the stated specificity of the primers, i.e., OHFV specific. Thus, we have to trust the authors on what they say in the text.
The authors state the emergence is “unprecedented” in the title but this is overstating the results as it is only based on qRT-PCR, no virus analysis, and if no one had looked in these areas before then I am not sure why it is unprecedented. There are many examples of viruses being discovered in areas not seen previously because no one has looked at samples in these areas. Thus, the authors need to explicitly state that no one has looked for OHFV in Kazakhstan previously. Lines 132-133 indicates this but it needs to be more explicit.
Line 85: “nucleolus”. I think the authors mean “nucleocapsid”?
Lines 213-217: Please state genome positions for the primers and the size of the amplicon.
References 3 and 8 are duplicates
Author Response
Reviewer 2:
This paper reports identification of Omsk hemorrhagic fever virus (OHFV) outside Siberia in Kazakhstan. This is a potentially important paper as it describes an expanded geographic range for the virus based on identification of OHFV in human, ticks and rodents. However, it is all based on qRT-PCR and no sequencing of PCR products, nor virus isolation. This would be particularly important as it would provide a second method to confirm the PCR results and also potentially look at genetic variation versus OHFV in Russia. To rely on one technique and make big conclusions has difficulties. Thus, at the very least a PCR product should be sequenced to confirm it is OHFV and compared to published OHFV strains.
Our response: Thank you to the reviewer for pointing out this major weakness of our study. As already stated in the remark of reviewer 1 we are at this moment unfortunately not able to perform sequencing in Kazakhstan due to the current political and pandemic situation. Export of biological material is not legal and so our efforts in this direction where unfortunately blocked. Still we would love to publish our initial data, albeit we are aware that they are far from epidemiological research standard, since we want to highlight the importance of research in Kazakhstan in the direction of OHFV.
The qRT-PCR is of concern as it is based on published primers from a book article not an article in a journal, so it is not easy to access it. I could not get hold of the article. The size of the amplicon is not stated and none of the control RT-PCR results are shown to demonstrate the stated specificity of the primers, i.e., OHFV specific. Thus, we have to trust the authors on what they say in the text.
Our response: Thank you for pointing out this issue. We are aware, that the primers were only published in a book and interestingly the author of the book chapter never used the primers in his own work. For the detection of OHFV usually a pan-flavivirus primer combination is employed and then the virus is identified by sequencing. Since sequencing is no longer an option for us we had to shift to alternative primers. This is how we came along this primer pair. We took those primers and the probe and checked for its efficacy. Here we saw that the primer OHF-S1F binds in OHFV E Gen (for example: X66694 and AF482341) at position 582-605, Primer OHF-d2R binds at 694 to 720, the OHF-TaqProbe binds at 599 to 626, leading to a 138bp fragment. Please see in Figure 1 an alignment of primers and probe.
Figure 1: Primer and Taq-probe sequence alignment to published OHFV sequences (E-gene)
To further confirm the specificity of the primers and probe we first performed an in silico analysis of the primer using NCBI blast to search for published sequences that fit to the sequence of the primers. As shown in Figure 2 for the forward primer only OHFV is identified with a 100% sequence identity. This is the same for the reverse primer and the probe.
Figure 2: Blast of the forward primer used in this study. Only OHFV sequences popped up in the query
Since we know, that in silico results and the real life sometimes do not go along we also tested this primer combination with our in house specification panel on flaviviruses. We also included other more distantly related viruses to get a convincing picture of the specificity of the primers. In Figure 3 you see the original qPCR result of the first run we performed, this is the basis for the data shown in the supplement.
Figure 3: Original data of the qPCR runs for the establishment of the specification panel of the OHFV primers and Taq-probe. Primers and probe were tested on isolates from closely related flaviviruses and others.
The authors state the emergence is “unprecedented” in the title but this is overstating the results as it is only based on qRT-PCR, no virus analysis, and if no one had looked in these areas before then I am not sure why it is unprecedented. There are many examples of viruses being discovered in areas not seen previously because no one has looked at samples in these areas. Thus, the authors need to explicitly state that no one has looked for OHFV in Kazakhstan previously. Lines 132-133 indicates this but it needs to be more explicit.
Our response: Thank you for this remark. This bold approach by us was also highlighted by reviewer 1. We toned down the statement a little and now only mention “first indications”, keeping in mind, that we are aware, that due to the lacking sequences we need to be careful of the soundness of our statement.
Indeed in Kazakhstan so far there were no previous attempts to screen for OHFV in the Russia neighbouring regions so far but we are optimistic that the responsible sanitary stations will due to this publication take the virus on their target list and initiate studies.
Line 85: “nucleolus”. I think the authors mean “nucleocapsid”?
Our response: Thank you for pointing out this error, indeed we meant the nucleocapsid.
Lines 213-217: Please state genome positions for the primers and the size of the amplicon.
Our response: Thank you for pointing out this missing information. We initially found it an unimportant information, since we only loaded the resulting fragments on a gel to prove the effectiveness of the qPCR run. Now we added the sentence “The primers bind the region from position 582 to position 720 and produce a fragment of 138 bp in length.”
References 3 and 8 are duplicates
Our response: Thank you for pointing out this duplication. We cleared the duplicate and also revisited the rest of the references and completed information where appropriate.

Round 2
Reviewer 1 Report
The authors addressed all my points in the revised version of the manuscript.
Author Response
Our response: Thank you so much for the support of our submission and your time to read and improve our paper
Reviewer 2 Report
This is an improved version of the manuscript. My only suggestion is to add the strains used for each virus in supplement #3.
Author Response
Our response: Thank you so much for reading again our manuscript and deeming it now as improved. We happily provided strain information for most of the viruses utilized. However, we have to admit, that our virus collection grew over the years and for some viruses there is only incomplete strain- or sequenceinformation available. In the course of the last years we initiated an entire sequencing of all our viruses in the database, however this project got massively delayed due to the occupation of the sequence facility the by coronavirus pandemic. But the identity of all the viruses is confirmed by either serological or molecular biological methods.